# MSSA-DEED: A Multi-Objective Salp Swarm Algorithm for Solving Dynamic Economic Emission Dispatch Problems

**Mohamed H. Hassan** [1], **Salah Kamel** [1], **José Luís Domínguez-García** [2] **and Mohamed F. El-Naggar** [3,4,*]

1. Department of Electrical Engineering, Faculty of Engineering, Aswan University, Aswan 81542, Egypt
2. Institut de Recerca en Energia de Catalunya (IREC), 08930 Sant Adriàdel Besos, Spain
3. Department of Electrical Engineering, College of Engineering, Prince Sattam Bin Abdulaziz University, Al-Kharj 16273, Saudi Arabia
4. Department of Electrical Power and Machines Engineering, Faculty of Engineering, Helwan University, Helwan 11795, Egypt
* Correspondence: mfelnaggar@yahoo.com

**Abstract:** Due to the rising cost of fuel, increased demand for energy, and the stresses of environmental issues, dynamic economic emission dispatch (DEED), which is the most precise mode for actual dispatching conditions, has been a significant study topic in current years. In this article, the higher dimensional, deeply correlated, non-convex, and non-linear multi-objective DEED problem is designated, involving both the fuel costs and emissions objectives simultaneously. In addition, the valve point effect, transmission loss, as well as the ramping rate, are considered. The Salp Swarm Algorithm (SSA) is a well-established meta-heuristic that was inspired by the foraging behavior of salps in deep oceans and has proved to be beneficial in estimating the global optima for many optimization problems. The objective of this article is to evaluate the performance of the multi-objective Salp Swarm Algorithm (MSSA) for obtaining the optimal dispatching schemes. Furthermore, the fuzzy decision-making (FDM) approach is employed to achieve the best compromise solution (BCS). In order to confirm the efficacy of the MSSA, the IEEE 30-bus six-unit power system, standard 39-bus ten-unit New England power system, and IEEE 118-bus fourteen-unit power system were considered as three studied cases. The obtained results proved the strength and supremacy of the MSSA compared with two well-known algorithms, the multi-objective grasshopper optimization algorithm (MOGOA) and the multi-objective ant lion optimizer (MALO), and other reported methods. The BCS of the proposed MSSA for the six-unit power system was USD 25,727.57 and 5.94564 Ib, while the BCS was $2.520778 \times$ USD 106 and $3.05994 \times 105$ lb for the ten-unit power system, and was $1.29200 \times$ USD106 and 98.1415 Ib for the 14 generating units. Comparisons with the other well-known methods revealed the superiority of the proposed MSSA and confirmed its potential for solving other power systems' multi-objective optimization problems.

**Keywords:** dynamic economic emission dispatch; MSSA; multi-objective salp swarm algorithm; greenhouse gases; valve point loading

## 1. Introduction

The need for energy is growing with the continued progress of industrial development on a world scale [1]. Adaptively modifying the power supply approach based on variations in load to enhance the economy of the power system has been a critical problem that must be solved in the optimization of the power system operation [2]. The EED that improves both the fuel cost and emissions according to the conventional ELD has gradually become a significant method to decrease the emission level of electric power systems [3]. The EED aims to adjust the approaches of the current power dispatch compared with developing power generation equipment or increasingly expanding the renewable energy resource [4], and extensive effort has newly been executed on the EED problem [5–7]. Nonetheless, EED studies specifically concentrate on the static model that does not take into consideration

the relationship between various interval periods for dispatching. As an extension of the EED, the DEED, which is the best mode for real dispatching conditions, has acquired more academic concern [8], despite the DEED issue being a highly dimensional, deeply correlated, non-convex, and non-linear MOP taking into consideration the incompatible objectives and the operational limitations. When there are transmission losses, the changing of the power demand and the POZ are also involved, and the DEED model becomes more complex.

Due to the complicatedness and significance of the DEED issue, it is completely essential to select a suitable model of dispatch and to choose proficient algorithms to acquire the best dispatching scheduling for the power generation. Recently, meta-heuristic approaches have become extremely popular for achieving the optimum solutions for numerous engineering problems [9], including the estimation of PV parameters [10], optimal power flow incorporating renewable energy and FACTs devices [11–13], and optimum reactive power dispatch with renewable energy uncertainty [14]. This popularity is because of many principal causes: a gradient-free mechanism, flexibility, and the local optima avoidance of these techniques [15].

Some studies have been published to handle the DEED problem. At first, the atmospheric pollutants were not taken into consideration as an objective in the DEED. In [16], a DEED problem was constructed without the ramp rate bounds of the power generators, the emission, and the security constraints, to the DEED model. Over the recent years, with the development of highly effective and more precise solving techniques, the research path of the DEED has changed to consider fuel cost and atmospheric pollutants simultaneously as multi-objectives. In [17], Basu employed an EPFSA to find the optimum solution for the this problem as a single-objective problem. In [18–20], the DEED problem was transformed to a single-objective problem based on the price penalty factor method and achieved the best solution for it by the IBFA, the neural network, and the ODEA. In [21–23], single-objective DEED models were solved using the PSO-based goal-attainment method, fuzzy adaptive modified theta-PSO, and hybrid DE- SQP, respectively. All of the research mentioned above have attained acceptable results. Nevertheless, because of the boundary of the DEED problem as a single-objective problem, it is challenging to acquire more than one optimum solution in an individual trial; additionally, the Pareto Front is surely difficult to achieve. In [24–28], the NSGA-II, the MODE, AMODE, the Chemical Reaction Optimization (CRO), and improved NSGA-II techniques were proposed to achieve the solution for the multi-objective DEED problem with competing and non-commensurable objectives. Additionally, in [29], the GSOMP technique was proposed and the DEED was formulated as a group of static economic emission dispatches based on the period intervals of dispatching that would have intricacies in the grouping of the best solutions from various time intervals. An improved tunicate swarm algorithm (ITSA) [30] and an improved PSO algorithm with a clone selection (PSOCS) [31] were improved to reach the optimal solution for the DEED problem. Multi-objective hybrid differential evolution with the simulated annealing technique (MOHDE-SAT) was selected as the best compromise solution for the DEED problem in [32]. A new approach to separating the DEED problem into offline training steps and online inference steps was proposed in [33]. Multi-objective Neural Networks trained using Differential Evolution (MONNDE) was proposed in [34] to build a function approximator that, once trained, could be used to predict the optimal solution for any change in load.

Recently, DEED has come to be the essential problem in the dynamic dispatch field of both micro-grids and smart grids. For such complex MOPs, it would permanently be the effort of the literature to use a high-performance technique that achieves more accurate solutions based on the particular characteristics of the problems. In recent decades, the optimization algorithm has received extensive attention and applications. The MSSA is a swarm intelligence-based multi-objective optimization algorithm that has been recently developed in [15]. The MSSA has been tested using benchmark functions and has been compared with other well-known algorithms [12]. It is an extension of the Salp Swarm Algorithm (SSA). Therefore, the basic theory of the SSA is firstly presented, and then the

principles of the MSSA are introduced. The main contributions of the paper are summarized as follows:

- Proposing the MSSA to find the optimal solution of the non-linear and non-convex multi-objective DEED problems.
- Solving multi-objective DEED problems considering the valve point effect, transmission loss, as well as the ramping rate.
- Applying the fuzzy decision-making approach to achieve the best compromise solutions.
- Using the 6-unit power system, 10-unit power system, and 14-unit power system as three studied cases to prove the efficacy of the MSSA.
- The obtained results prove the strength and supremacy of the MSSA in comparison with two well-known algorithms, the MALO [35] and MOGOA [36], and other reported methods.

The remainder of the article is arranged as follows. Section 2 presents the mathematical model of the DEED problem. The suggested MSSA is shown in Section 3. The simulation results and discussions are given in Section 4. Finally, Section 5 presents the conclusion.

## 2. Mathematical Model of EED Problem

### 2.1. Objective Functions

The fuel cost of each generator taken into consideration, the VPE, is formulated as the summation of a quadratic and a sinusoidal function. Figure 1 shows a VPE on the fuel cost function [37]. Therefore, the fuel cost of $N_G$ generators through $T$ dispatching period intervals is given from the following equation [38]:

$$F_1 = \sum_{t=1}^{T} \sum_{i=1}^{N_G} \left[ a_i + b_i P_{Gi,t} + c_i P_{Gi,t}^2 + \left| d_i \sin \left( e_i \left( P_{Gi}^{min} - P_{Gi,t} \right) \right) \right| \right] \tag{1}$$

where $a_i$, $b_i$, $c_i$, $d_i$, and $e_i$ denote the cost coefficients for the $i$th unit; $P_{Gi,t}$ refers to the power output of the $i$th ($i$ = 1; 2; 3; ... ; $N_G$) unit at the dispatching time interval $t$; and $N_G$ is the number of generating units.

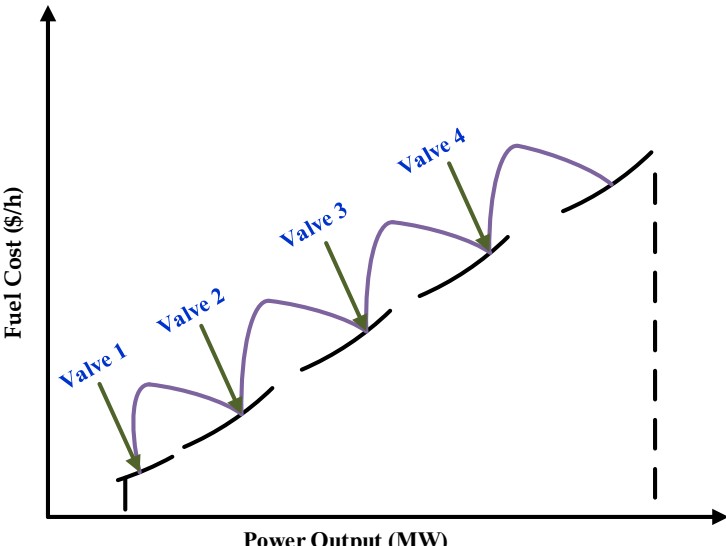

**Figure 1.** VPE on a fuel cost function.

Commonly, atmospheric pollutants including $CO_2$, $SO_x$, and $NO_x$ produced by fossil-fueled thermal generators are mathematically represented as the totality of a quadratic and

an exponential function. Accordingly, the total atmospheric pollutants of $N_G$ generators through the dispatching interval is expressed as [39]:

$$F_2 = \sum_{t=1}^{T} \sum_{i=1}^{N_G} \left[ \alpha_i + \beta_i P_{Gi,t} + \gamma_i P_{Gi,t}^2 + \eta_i \exp(\delta_i P_{Gi,t}) \right] \tag{2}$$

where $\alpha_i$, $\beta_i$, $\gamma_i$, $\eta_i$, and $\delta_i$ denote the emission coefficients for the $i$th unit.

### 2.2. Operational Constraints

#### 2.2.1. Power Balance Constraint

The total power generated should equal the summation of the total power demand $P_{D,t}$ and the complete transmission losses $P_{L,t}$, which is formulated as:

$$\sum_{i=1}^{n} P_{i,t} - P_{D,t} - P_{L,t} = 0 \tag{3}$$

where $P_{L,t}$ is calculated by Kron's formula:

$$P_{L,t} = \sum_{i=1}^{n} \sum_{j=1}^{n} P_{i,t} B_{ij} P_{j,t} + \sum_{i=1}^{n} B_{i0} P_{i,t} + B_{00} \tag{4}$$

where $B_{ij}$, $B_{i0}$, and $B_{00}$ are the $B$-matrix coefficients for $P_{L,t}$.

#### 2.2.2. Generating Capacity Constraint

The power output $P_{i,t}$ must be within the minimum and maximum power generation bounds, as represented by:

$$P_i^{min} \leq P_{i,t} \leq P_i^{max} \tag{5}$$

where $P_i^{max}$ is the maximum limit of the $i$th generator.

#### 2.2.3. Ramp Rate Constraint

Under real-world conditions, the operating limit of each generator is bounded by its ramp rate bound; therefore, the output power $P_i$ cannot be adjusted immediately. The up and down ramp limits are represented by [40]:

$$\begin{cases} P_{i,t} - P_{i,t-1} - UR_i \times \Delta T \leq 0 \\ P_{i,t-1} - P_{i,t} - DR_i \times \Delta T \leq 0 \end{cases} \tag{6}$$

where $UR_i$ and $DR_i$ are the up and down limit of generator $i$, respectively. $\Delta T$ denotes the length of each dispatching time interval.

## 3. Multi-Objective Optimization Algorithm

The multi-objective technique has been employed to achieve the solution for several MOPs. A recent multi-objective technique that is called the MSSA is presented in this paper.

### 3.1. Multi-Objective Optimization Problem

The DEED model is represented as a MOP with non-linear constraints based on the objective functions and system constraints as follows:

$$\begin{cases} \quad \min \quad [F_1(P), F_2(P)] \\ s.t. \quad\quad g_i(P) = 0, \quad k = 1, 2, \ldots, u \\ \quad\quad h_i(P) \leq 0, \quad\quad i = 1, 2, \ldots, q \end{cases} \tag{7}$$

where $F_1(P)$ and $F_2(P)$ refer to the fuel cost and emission objective functions, respectively, $g_i(P)$ and $h_i(P)$ denote the equality and inequality restrictions, and $u$ and $q$ represent the

number of corresponding formulations, respectively. *P* refers to the active power decision control vector of the generators.

With a single objective, it can be confidently assessed that a solution is superior to another by relying on comparing the single criterion, whereas in an MOP, there is more than a single criterion to check and compare the obtained solutions. Pareto optimal (PO) dominance is the principal approach that is used to compare two solutions for multiple objectives and it is described in [41]. A priori and a posteriori are the two principal methods to achieve the best solution for the MOPs. In the prior approach, the MOP is converted into a single-objective problem by the summation of the objectives with a group of weights specified by experts. The major shortcoming of this approach is that the Pareto optimal collection and the front need to be constructed by re-running the technique and modifying the weights. Nevertheless, the posteriori approach maintains the multi-objective construction in the solving strategy, and the Pareto optimal set can be chosen in a single run. Without any weight to be explained by experts, this method can estimate any Pareto Front type. Due to the posterior optimization benefits over the a priori method, the goal of our study is targeted at a posteriori multi-objective optimization.

*3.2. Multi-Objective Salp Swarm Algorithm*

Salps are a kind of marine organism from the Salpidae family, with a comparable appearance to jellyfish. In the strategy of foraging food, these salps display a swarm performance, establishing a salp chain that stimulates the SSA technique suggested in [15]. The population in a salp chain contains a leader and a set of followers, where the leader tries to find the source of the food while the followers vary their location according to the salp ahead of them, and accordingly to the leader. Regarding an optimization problem with $n$ variables and the $x_i$ location of the salp $i$, denoted by a vector of $n$ elements $x_i = [ x_j^1, x_j^2, \ldots, x_j^n ]$, the leader's location in the salp chain updates through this equation:

$$x_j^1 = \begin{cases} F_j + c_1\big((ub_j - lb_j)\big)c_2 + lb_j, c_3 > 0.5 \\ F_j - c_1\big((ub_j - lb_j)\big)c_2 + lb_j, c_3 \leq 0.5 \end{cases} \qquad (8)$$

where $F_j$ denotes the value in the $j$th dimension of the food source $F$ (the best location), $ub_j$ and $lb_j$ are the upper and lower bounds, respectively, in the $j$th dimension, and the parameters $c_2$ and $c_3$ represent randomly produced numbers in the interval [0, 1]. The parameter $c_1$ can be calculated from the equation below:

$$c_1 = 2e^{-\left(\frac{4t}{T}\right)^2} \qquad (9)$$

where $t$ denotes the present iteration while $T$ represents the maximum number of iterations.

When the leader's location updated, the followers' location is varied by the following equation:

$$x_j^i = \frac{1}{2}\left(x_j^i + x_j^{i-j}\right) \qquad (10)$$

where $x_j^i$ refers to the location in $j$th dimension of agent $i$, with $2 \leq i \leq n$.

The swarm behavior of salp chains is simulated based on the mathematical equations explained above. When dealing with MOPs, two issues need to be modified for the SSA technique. Firstly, the MSSA needs to store many results as the best solutions for a MOP. Secondly, in each iteration, the SSA updates the food source with the optimal solution, but in the MOP, single optimal solutions are not existent.

In the MSSA, the primary problem is resolved by supplying the SSA technique with a food source repository. This repository can collect a determinate number of ND solutions. In the optimization operation, the Pareto dominance operators are used to compare each salp with all the residents in the repository. When a salp dominates only an agent in the repository, it needs to remove the exchanged salp. In another case, when a salp dominates a group of agents in the repository, they all must be from the repository and the salp must

be added to the repository. If at minimum one of the residents dominates a salp in the newly generated swarm, we should discard it directly. When a salp is non-dominated compared to the entire residents, it should be added to the archive. When the repository is full, it needs to reject one of the similar ND salps in the repository. For addressing the other problem, a suitable method is to choose it from a group of ND salps with the least crowded neighborhood. This can be performed by the same ranking method and RWS. The solution procedure of the MSSA for solving the DEED problem is depicted using the flowchart in Figure 2. Additionally, Algorithm 1 presents the pseudo code of the MSSA:

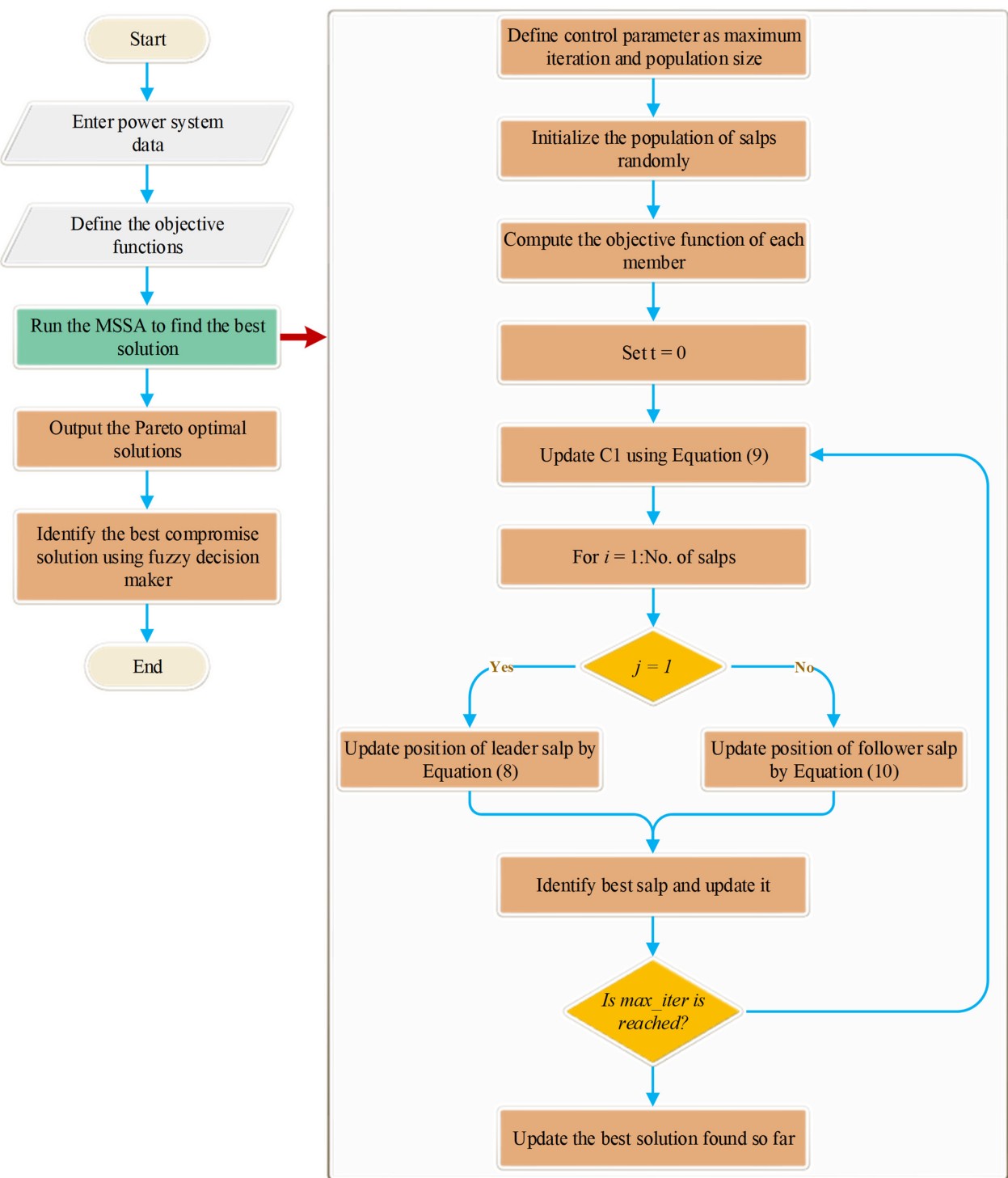

**Figure 2.** Flowchart for solution of DEED problem by MSSA.

---

**Algorithm 1** Pseudo code of MSSA

---

Initialize the parameters *the number of salps, Obj_no, dim, lb, ub, Archive Maxsize, max_iter*
Initialize the salp population $x_i (i = 1, 2, \ldots, n)$;
Define the objective function (Dynamic Economic Emission Dispatch function)
**While** ($t \leq max\_iter$)
    Evaluate the fitness of all salp with Ob_func;
    Find the ND solutions
    Update the repository in regard to the achieved ND agents
    *If* (repository is full);
        Perform the repository maintenance process to eliminate one repository neighborhood
        Insert the ND agent to the repository
    *End If*
    Select a source of food from repository
    Update $c_1$ by **Equation** (9);
    **For** *each search agent*
**If** (i==1)
        Update the location of the leading salp by **Equation** (8);
    *Else*
        Update the location of the leading salp by **Equation** (10);
*End if*
    **End For**
    $t = t + 1$;
**End While**
**Return** repository

---

## 4. Simulation Results and Discussions

In this section, the MSSA is applied to three test systems (six-generator, ten-generator, and fourteen-generator). The results attained by the MSSA are compared to the results of the MALO and MOGOA algorithms and those reported in previous articles. When accessing the experimental results, besides using the best fuel cost and best emission solutions, the BCS that is determined using a fuzzy-based mechanism is used [42]. To find the BCS, the membership value of each individual in the Pareto optimal set $F_i$ is calculated by the membership function as follows:

$$\mu_i = \begin{cases} 1 & F_i \leq F_i^{min} \\ \frac{F_i^{max} - F_i}{F_i^{max} - F_i^{min}} & F_i^{min} < F_i < F_i^{max} \\ 0 & F_i \leq F_i^{max} \end{cases} \tag{11}$$

where $F_i^{max}$ and $F_i^{min}$ are the maximum and minimum values of $F_i$ between all ND solutions, respectively.

The normalized membership function ($\mu^k$) is given by:

$$\mu^k = \frac{\sum_{i=1}^{Nobj} \mu_i^k}{\sum_{k=1}^{M} \sum_{i=1}^{Nobj} \mu_i^k} \tag{12}$$

where M denotes the total number of ND solutions. The BCS is chosen from all ND solutions based on the value of $\mu^k$, where it has a maximum value of $\mu^k$.

According to these three test systems, the three cases studied are tabulated in Table 1. The total scheduling period was 24 h with 1 h intervals and the load per hour stayed constant. In the simulations run of the three cases, each algorithm was implemented on a laptop with Core(TM) i5-4210U CPU (2.40 GHz), RAM 8 GB, and Windows 8.1 64-bit operating system, through MATLAB R2016a software. The IEEE 30-bus six-generator (Case 1) test system is displayed in Figure 3. The load demands of this test system are provided in Appendix A Table A1. The fuel cost and emission coefficients values for the IEEE 30-bus six-generator test system are provided in Appendix A Table A2.

**Table 1.** List of three studied cases.

| Case Number | Number of Generators | Number of Buses | Number of Decision Variables | Number of Equality Constraints |
|---|---|---|---|---|
| Case 1 | 6 | 30 | $6 \times 24 = 144$ | 24 |
| Case 2 | 10 | NA | $10 \times 24 = 240$ | 24 |
| Case 3 | 14 | 118 | $14 \times 24 = 336$ | 24 |

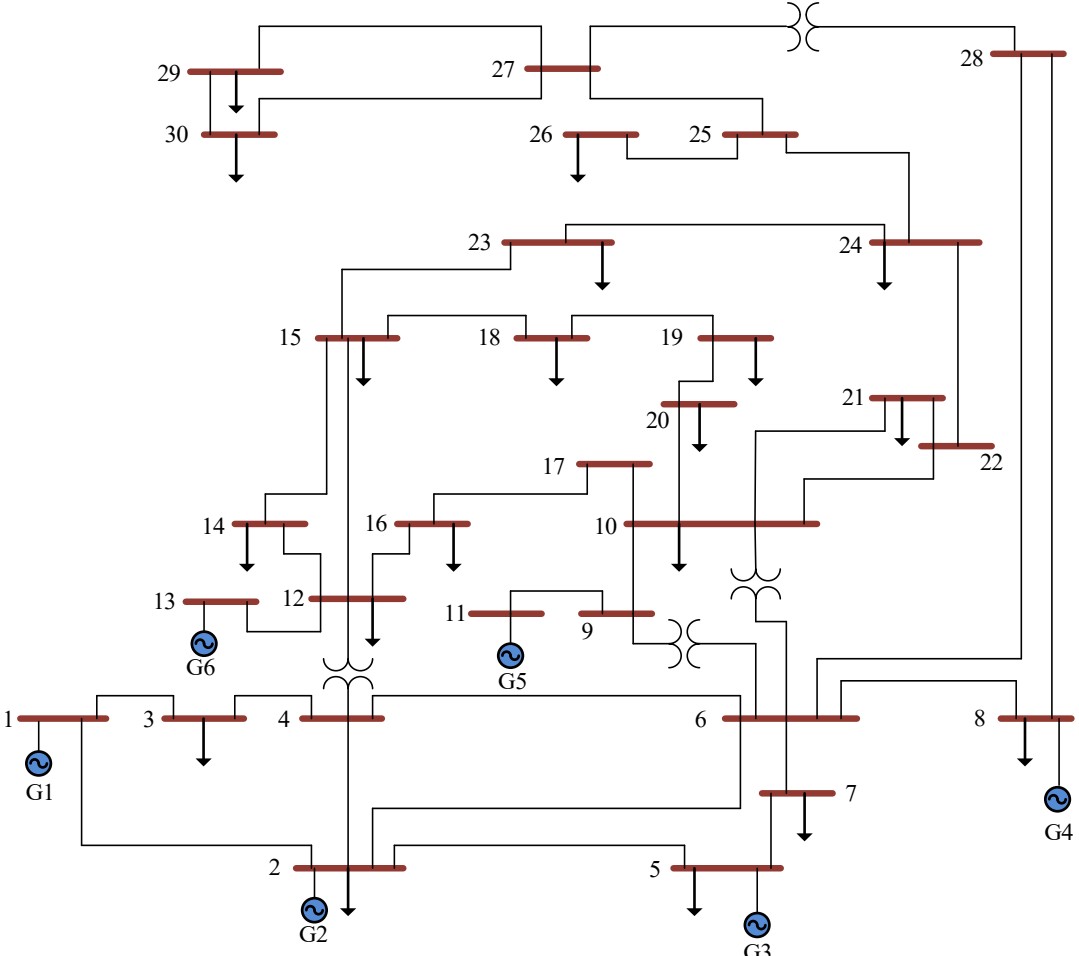

**Figure 3.** IEEE 30-bus six-generator test system.

The single line diagram of the 10-generator (Case 2) test system is displayed in Figure 4. This test system has 39 buses, 46 lines, and 10 thermal generation units. The power demands of this system are tabulated in Appendix B Table A3. The data pertaining to the thermal power generating units such as fuel cost coefficients, emission coefficients, the output power bounds, and ramp rate boundaries are shown in Appendix B Tables A4 and A5. Appendix B Table A6 presents the B-coefficients.

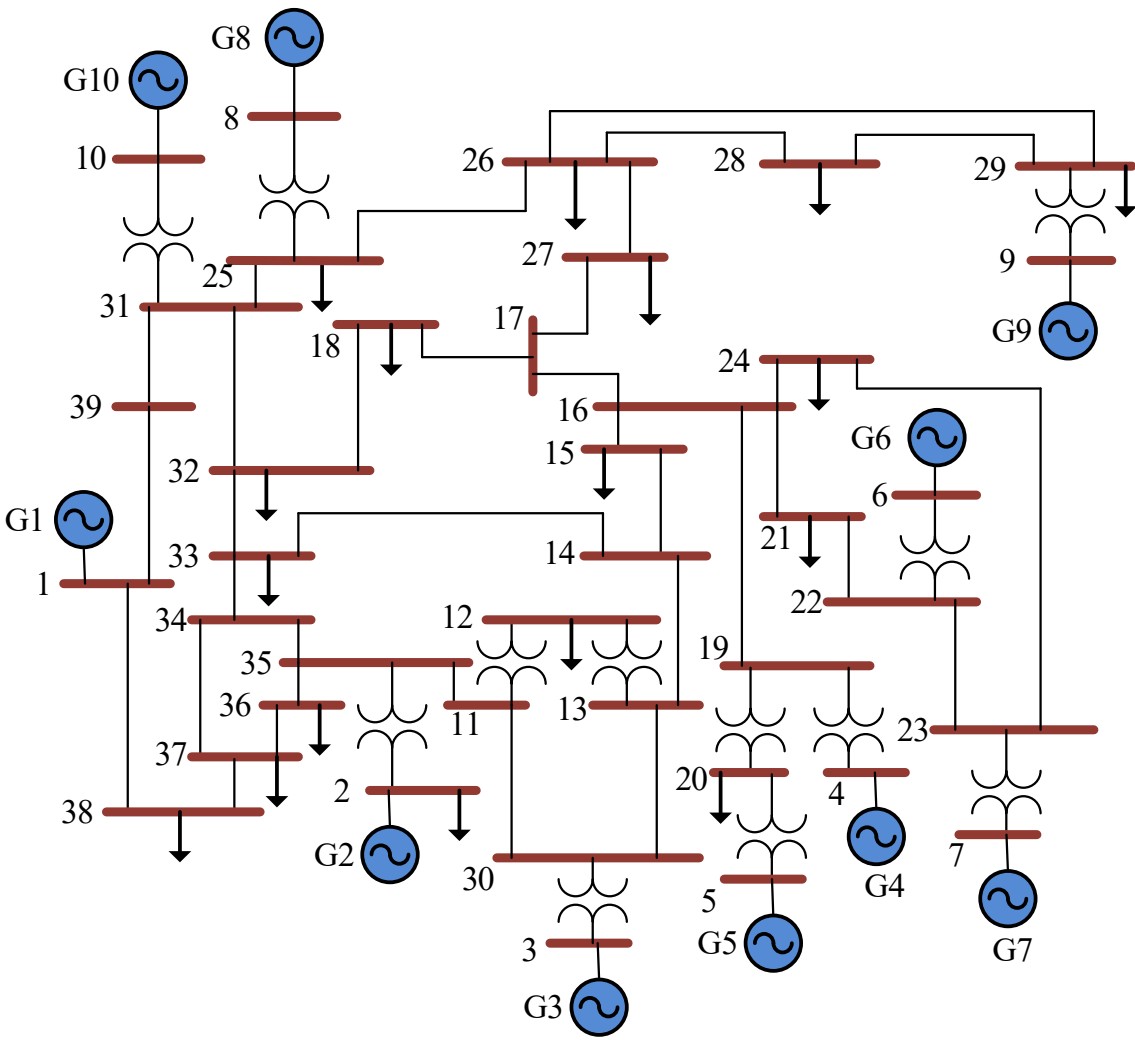

**Figure 4.** The 10-unit test system.

Appendix C Table A7 presents the load demands of the IEEE 118-bus 14-generator (Case 3) test system. The values of the fuel and emission coefficients of these case are given in Appendix C Table A8.

### 4.1. Case 1: Six-Unit Test System

In this case, the IEEE 30-bus six-unit power system was solved using the proposed algorithm. The non-smooth fuel cost function and power loss were considered in the DEED model. This system was tested and the Pareto Front and best compromise solution obtained by the proposed MSSA, MALO, and MOGOA algorithms are shown in Figure 5a–c, respectively. It can be clearly seen from these figures that the solutions obtained by the MSSA were better in convergence and diversity than those of the other algorithms. In addition, the best compromise solution of the other algorithms such as MAMODE [26] and GSOMP [29] are displayed in Figure 5d to demonstrate the effectiveness of the proposed MSSA in achieving the optimal solution for the DEED problem.

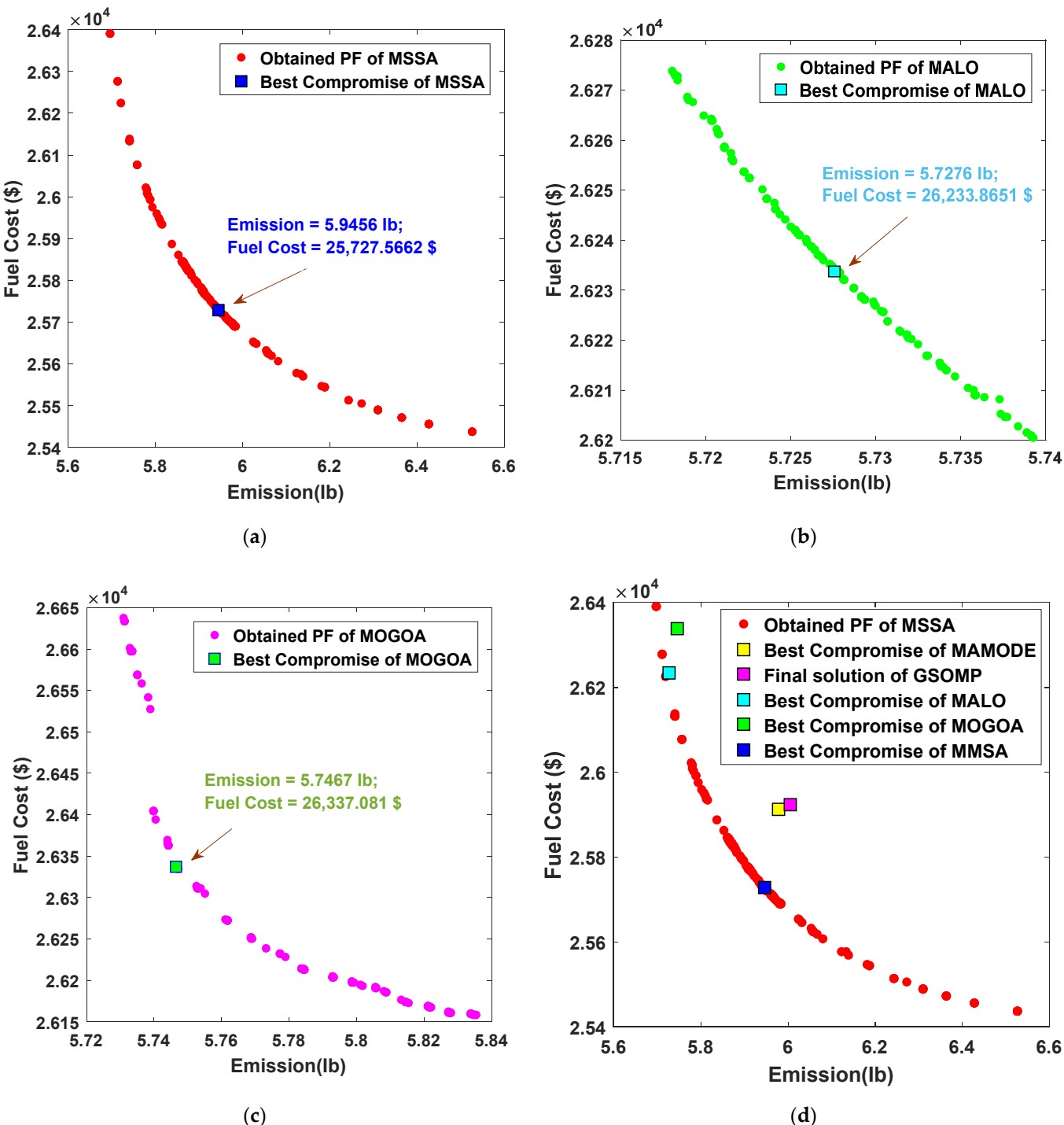

**Figure 5.** Pareto Front of the first case with power loss obtained by (**a**) MSSA; (**b**) MALO; (**c**) MOGOA; (**d**) MSSA and comparison of optimal solutions.

For comparison, the results of the BCS achieved by the MSSA, MALO, MOGOA and MAMODE [26], GSOMP [29], MOPSO [29], and NSGA-II [29] are listed in Table 2. From Table 2, it can be observed that the results obtained by the proposed MSSA were clearly more environmental and cost efficient, which demonstrates the effectiveness and superiority of the MSSA over all the other algorithms.

**Table 2.** The optimum objective values of Case 1.

| Algorithm | Objectives | Total Cost (USD) | Emission (Ib) |
|---|---|---|---|
| MSSA | BC | 25,437.59 | 6.52773 |
| | BE | 26,389.76 | 5.69654 |
| | BCS | **25,727.57** | **5.94564** |
| MALO | BC | 26,200.57 | 5.73927 |
| | BE | 26,273.83 | 5.71806 |
| | BCS | 26,233.87 | 5.72757 |
| MOGOA | BC | 26,158.07 | 5.83543 |
| | BE | 26,637.37 | 5.73097 |
| | BCS | 26,337.08 | 5.74666 |
| MAMODE | BC | 25,732 | NA |
| | BE | NA | 5.7283 |
| | BCS | 25,912.89 | 5.97955 |
| GSOMP | BC | 25,493 | NA |
| | BE | NA | 5.6847 |
| | BCS | 25,924.46 | 6.00415 |
| MOPSO | BC | 25,633.2 | NA |
| | BE | NA | 5.6863 |
| NSGA-II | BC | 25,507.4 | NA |
| | BE | NA | 5.6881 |

Bold values have the best performance; best cost (BC); best emission (BE); best compromise solution (BCS); not available (NA).

From the obtained results presented in Table 3, the power balance constraints were verified according to the complete information of the BCS. At each time interval, the sum of the output of the generating units matched the power load plus the power loss mutually, meaning the solution did not violate the equality constraints at each interval. The power balance constraints checking for the best compromise solution are shown in Figure 6. From the above result comparisons, it can be concluded that, compared with the other comparative methods, the proposed MSSA can provide better results and the effectiveness of the proposed technique for solving the DEED problem was verified.

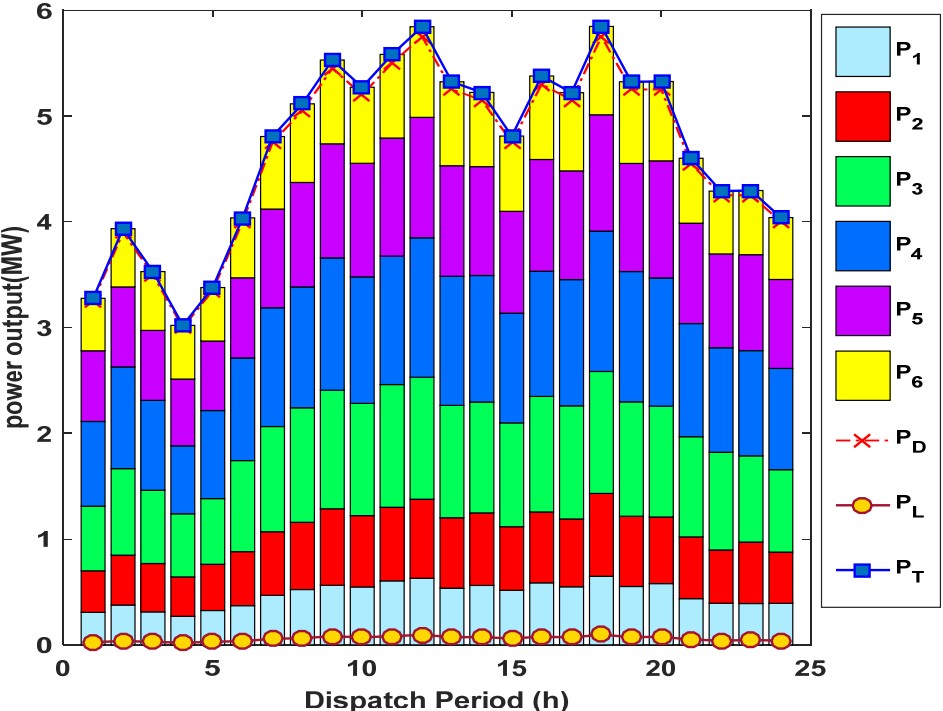

**Figure 6.** Constraints check for the BCS of Case 1 using MSSA.

**Table 3.** The BCS of the first case acquired using MSSA with power loss (p.u.).

| Hour | $P_1$ | $P_2$ | $P_3$ | $P_4$ | $P_5$ | $P_6$ | $P_T$ | $P_L$ |
|------|-------|-------|-------|-------|-------|-------|-------|-------|
| 1 | 0.3068 | 0.3926 | 0.6113 | 0.8005 | 0.6690 | 0.4950 | 3.2752 | 0.0252 |
| 2 | 0.3760 | 0.4722 | 0.8176 | 0.9616 | 0.7564 | 0.5511 | 3.9348 | 0.0348 |
| 3 | 0.3109 | 0.4573 | 0.6931 | 0.8493 | 0.6620 | 0.5560 | 3.5287 | 0.0287 |
| 4 | 0.2697 | 0.3719 | 0.5970 | 0.6426 | 0.6305 | 0.5080 | 3.0196 | 0.0196 |
| 5 | 0.3240 | 0.4369 | 0.6214 | 0.8330 | 0.6568 | 0.5060 | 3.3782 | 0.0282 |
| 6 | 0.3700 | 0.5111 | 0.8599 | 0.9713 | 0.7568 | 0.5666 | 4.0358 | 0.0358 |
| 7 | 0.4673 | 0.6018 | 0.9944 | 1.1233 | 0.9327 | 0.6861 | 4.8056 | 0.0556 |
| 8 | 0.5225 | 0.6361 | 1.0820 | 1.1433 | 0.9867 | 0.7439 | 5.1145 | 0.0645 |
| 9 | 0.5634 | 0.7228 | 1.1207 | 1.2508 | 1.0785 | 0.7930 | 5.5292 | 0.0792 |
| 10 | 0.5478 | 0.6738 | 1.0600 | 1.1967 | 1.0742 | 0.7190 | 5.2716 | 0.0716 |
| 11 | 0.6053 | 0.6953 | 1.1598 | 1.2153 | 1.1160 | 0.7898 | 5.5815 | 0.0815 |
| 12 | 0.6297 | 0.7469 | 1.1529 | 1.3184 | 1.1386 | 0.8572 | 5.8437 | 0.0937 |
| 13 | 0.5358 | 0.6647 | 1.0635 | 1.2200 | 1.0455 | 0.7936 | 5.3230 | 0.0730 |
| 14 | 0.5614 | 0.6853 | 1.0480 | 1.1968 | 1.0295 | 0.7012 | 5.2220 | 0.0720 |
| 15 | 0.5151 | 0.6013 | 0.9823 | 1.0354 | 0.9633 | 0.7117 | 4.8091 | 0.0591 |
| 16 | 0.5863 | 0.6697 | 1.0924 | 1.1834 | 1.0571 | 0.7879 | 5.3769 | 0.0769 |
| 17 | 0.5480 | 0.6413 | 1.0702 | 1.1935 | 1.0275 | 0.7392 | 5.2197 | 0.0697 |
| 18 | 0.6483 | 0.7825 | 1.1536 | 1.3271 | 1.0989 | 0.8356 | 5.8460 | 0.0960 |
| 19 | 0.5509 | 0.6653 | 1.0803 | 1.2327 | 1.0223 | 0.7717 | 5.3233 | 0.0733 |
| 20 | 0.5788 | 0.6293 | 1.0482 | 1.2115 | 1.1082 | 0.7498 | 5.3258 | 0.0758 |
| 21 | 0.4362 | 0.5833 | 0.9479 | 1.0702 | 0.9487 | 0.6131 | 4.5995 | 0.0495 |
| 22 | 0.3931 | 0.5030 | 0.9248 | 0.9880 | 0.8860 | 0.5946 | 4.2896 | 0.0396 |
| 23 | 0.3901 | 0.5804 | 0.8155 | 0.9958 | 0.9065 | 0.6071 | 4.2954 | 0.0454 |
| 24 | 0.3915 | 0.4850 | 0.7786 | 0.9582 | 0.8404 | 0.5858 | 4.0395 | 0.0395 |

*4.2. Case 2: 10-Unit Test System*

In this case, the standard 39-bus ten-unit New England power system with the consideration of power loss was tested. The problem had 24 non-linear equality constraints. Figure 7 presents the Pareto Front of case two with a transmission loss achieved using the proposed MSSA, MALO, MOGOA, and the BCS of other algorithms such as the RCGA, NSGA-II, and MODE. From the Pareto Front curves it can be seen that the proposed MSSA can give well-disseminated solutions with the fuel cost and emission functions. In addition, it is clear to see that the Pareto solutions are distributed uniformly in the objective space from Figure 7.

The dispatching results of Case 2 that were achieved using the proposed MSSA were compared with two well-known algorithms in detail, as displayed in Table 4. From this table, it is seen that the BCS of the MSSA was USD $2.520778 \times 10^6$ and $3.05994 \times 10^5$ lb, which is superior to the several compared techniques. In terms of the economy and environment, the cost and emission of the BCS of the MSSA technique was superior to the NSGA II [24], RCGA [24], MODE [25], MALO, and MOGOA. In general, the solution of the MSSA technique was superior to the comparison techniques.

The details of the BCS attained using the MSSA are shown in Table 5. At each interval, the summation of the generators' output equaled the summation of the load demand plus the transmission losses, meaning the solution did not violate the equality restrictions at each time interval. From these simulation results, it can be concluded that, compared with the other comparative techniques, the MSSA provided the best results, meaning the efficiency of the MSSA for solving the DEED problem was confirmed again. Figure 8 illustrate the power balance restrictions that verified the $P_T$, $P_D$, and $P_L$ at each interval.

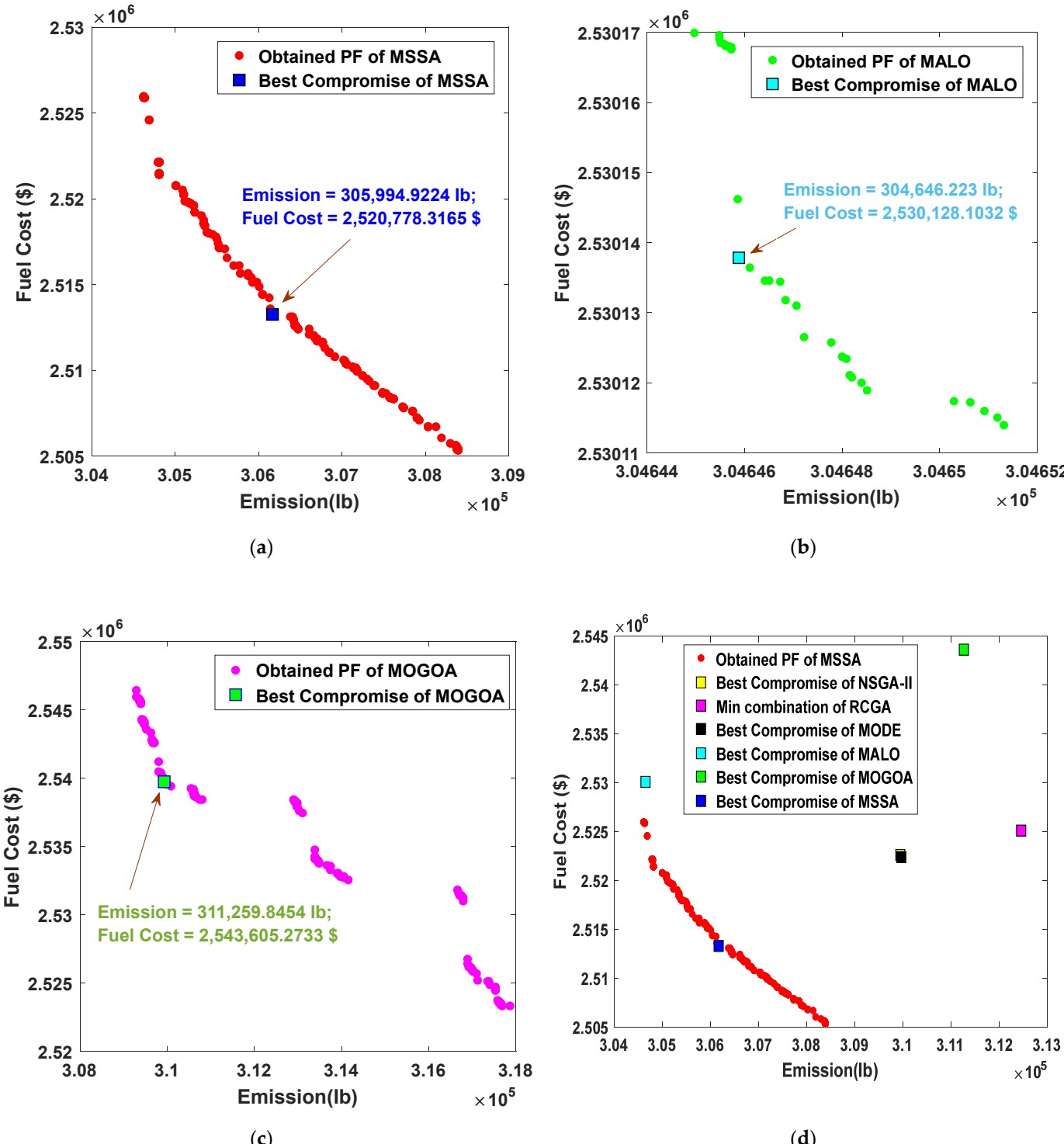

**Figure 7.** Pareto Front of Case 2 with power loss obtained by (**a**) MSSA; (**b**) MALO; (**c**) MOGOA; (**d**) MSSA and comparison of optimum solutions.

### 4.3. Case 3: 14-Unit Test System

In this case, the IEEE 118-bus system with 14 generating units considering the power loss of the network was studied to confirm its effectiveness in solving the high dimensional DEED problem with non-linear objectives and constraints. The best Pareto Front of case three with a power loss attained using the proposed MSSA, MALO, and MOGOA algorithms is shown in Figure 9. It is seen that the BCS of the proposed MSSA technique

was USD $1.292 \times 10^5$ and 98.1415 lb, which was better than the two compared techniques. Moreover, from this figure, we can see directly that the Pareto solutions of the proposed MSSA were widely and well distributed.

**Table 4.** The optimum objective values of the second case.

| Algorithm | Objectives | Total Cost/$10^6$ (USD) | Emission/$10^5$ (Ib) |
|---|---|---|---|
| MSSA | BC | 2.505334 | 3.083911 |
| | BE | 2.525936 | 3.046193 |
| | BCS | **2.520778** | **3.05994** |
| MALO | BC | 2.530114 | 3.0465133 |
| | BE | 2.5301699 | 3.0464497 |
| | BCS | 2.530128 | 3.04646 |
| MOGOA | BC | 2.5233373 | 3.178493 |
| | BE | 2.546446 | 3.0928948 |
| | BCS | 2.5436 | 3.1126 |
| NSGA-II | BC | 2.5168 | 3.1740 |
| | BE | 2.6563 | 3.0412 |
| | BCS | 2.5226 | 3.0994 |
| RCGA | BC | 2.5168 | 3.1740 |
| | BE | 2.6563 | 3.0412 |
| | BCS | 2.5251 | 3.1246 |
| MODE | BC | 2.5123 | 3.0113 |
| | BE | 2.5436 | 2.9607 |
| | BCS | 2.5224 | 3.0997 |

Bold values have the best performance; best cost (BC); best emission (BE); best compromise solution (BCS).

**Table 5.** The BCS of the second case acquired using MSSA with transmission losses (p.u.).

| Hour | $P_1$ | $P_2$ | $P_3$ | $P_4$ | $P_5$ | $P_6$ | $P_7$ | $P_8$ | $P_9$ | $P_{10}$ | $P_L$ |
|---|---|---|---|---|---|---|---|---|---|---|---|
| 1 | 150.01 | 138.07 | 166.79 | 132.28 | 153.54 | 117.74 | 57.83 | 64.50 | 54.36 | 20.46 | 19.59 |
| 2 | 150.09 | 142.01 | 135.31 | 167.76 | 148.41 | 134.97 | 70.44 | 74.86 | 65.90 | 42.62 | 22.38 |
| 3 | 151.11 | 149.39 | 186.96 | 177.14 | 160.71 | 129.73 | 96.10 | 102.74 | 79.65 | 53.16 | 28.68 |
| 4 | 180.36 | 173.72 | 188.94 | 189.97 | 170.87 | 159.04 | 125.90 | 118.29 | 79.96 | 55.00 | 36.05 |
| 5 | 152.47 | 191.60 | 185.68 | 227.73 | 219.00 | 159.49 | 129.98 | 119.07 | 79.83 | 54.92 | 39.78 |
| 6 | 216.24 | 195.52 | 216.39 | 264.98 | 241.20 | 157.90 | 129.89 | 119.96 | 79.93 | 54.92 | 48.92 |
| 7 | 212.53 | 234.47 | 271.75 | 254.13 | 238.67 | 159.30 | 130.00 | 120.00 | 79.96 | 55.00 | 53.81 |
| 8 | 230.46 | 237.72 | 303.67 | 277.65 | 242.83 | 159.54 | 129.50 | 119.32 | 79.66 | 54.57 | 58.92 |
| 9 | 283.42 | 303.83 | 325.48 | 295.22 | 242.08 | 160.00 | 129.92 | 119.88 | 79.99 | 54.99 | 70.80 |
| 10 | 332.92 | 342.01 | 338.70 | 299.98 | 243.00 | 160.00 | 130.00 | 120.00 | 79.99 | 54.99 | 79.58 |
| 11 | 376.74 | 389.20 | 339.96 | 300.00 | 243.00 | 160.00 | 129.99 | 120.00 | 80.00 | 55.00 | 87.88 |
| 12 | 397.87 | 416.60 | 339.99 | 300.00 | 243.00 | 160.00 | 130.00 | 120.00 | 80.00 | 55.00 | 92.46 |
| 13 | 356.03 | 372.61 | 339.88 | 299.98 | 242.96 | 159.99 | 130.00 | 119.99 | 79.99 | 55.00 | 84.44 |
| 14 | 292.50 | 296.25 | 318.86 | 299.34 | 242.96 | 159.97 | 129.99 | 119.98 | 79.99 | 54.98 | 70.82 |
| 15 | 230.56 | 229.74 | 292.52 | 294.79 | 242.63 | 160.00 | 129.58 | 120.00 | 80.00 | 55.00 | 58.82 |
| 16 | 171.31 | 174.43 | 237.51 | 245.09 | 240.74 | 157.37 | 124.65 | 118.51 | 75.72 | 52.61 | 43.94 |
| 17 | 157.28 | 150.24 | 203.43 | 235.55 | 238.54 | 155.80 | 127.08 | 119.30 | 77.52 | 54.81 | 39.54 |
| 18 | 213.11 | 203.73 | 246.62 | 245.80 | 226.31 | 158.13 | 129.67 | 119.72 | 79.40 | 54.44 | 48.94 |
| 19 | 232.58 | 239.14 | 295.31 | 280.12 | 242.93 | 159.98 | 129.98 | 119.92 | 79.96 | 55.00 | 58.93 |
| 20 | 309.10 | 311.05 | 338.90 | 300.00 | 243.00 | 159.86 | 130.00 | 120.00 | 80.00 | 55.00 | 74.90 |
| 21 | 286.57 | 288.13 | 334.04 | 297.93 | 243.00 | 160.00 | 130.00 | 120.00 | 80.00 | 55.00 | 70.67 |
| 22 | 211.19 | 217.82 | 260.45 | 248.14 | 225.32 | 146.40 | 119.67 | 116.17 | 78.09 | 53.94 | 49.19 |
| 23 | 151.66 | 139.84 | 197.33 | 208.20 | 202.88 | 104.38 | 129.89 | 117.26 | 71.11 | 41.56 | 32.11 |
| 24 | 151.96 | 137.93 | 164.64 | 182.20 | 198.39 | 109.62 | 102.80 | 90.20 | 42.34 | 29.29 | 25.38 |

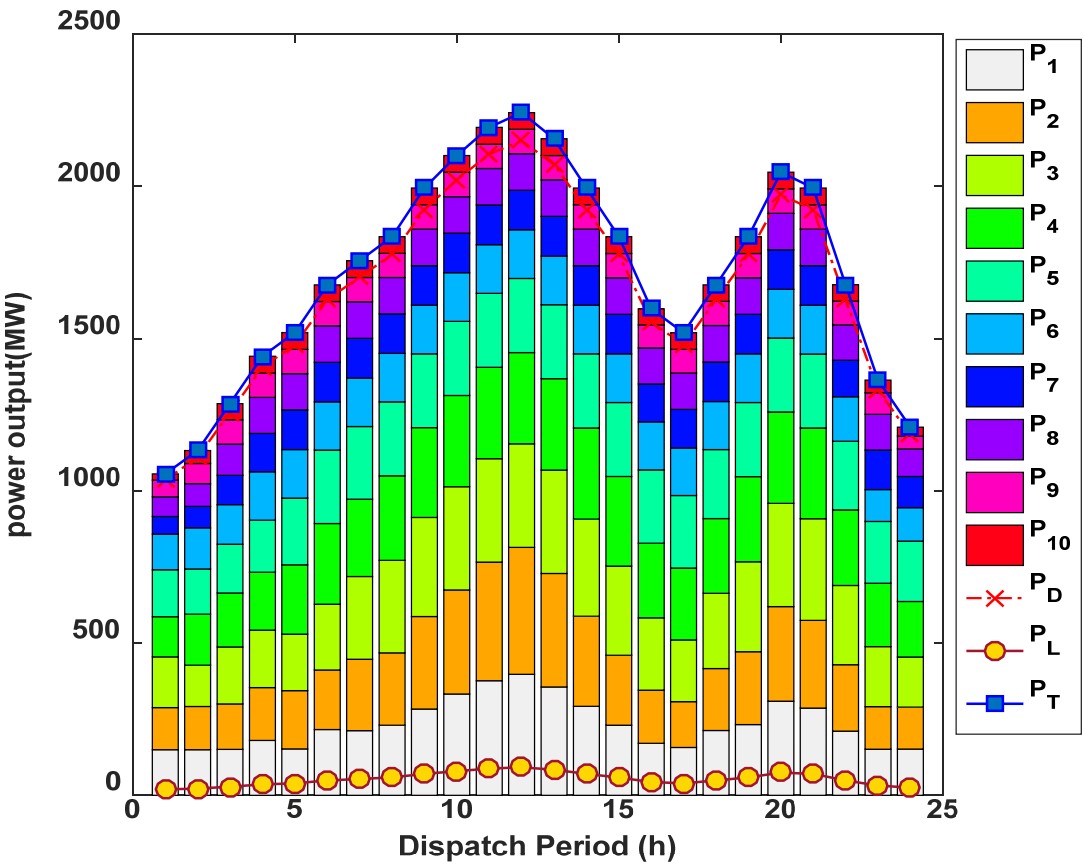

**Figure 8.** Constraints check for the BCS of the second case using MSSA.

The obtained results of the proposed technique were compared with others, which confirmed the capability of the proposed technique to find the optimal solution for the DEED problem. The detailed compared results are displayed in Table 6.

**Table 6.** Comparison of the optimum objective values of Case 3.

| Algorithm | Objectives | Total Cost//$10^5$ (USD) | Emission (Ib) |
|---|---|---|---|
| MSSA | BC | 1.2548728 | 110.02359 |
| | BE | 1.3560793 | 91.69988 |
| | BCS | **1.29200** | **98.1415** |
| MALO | BC | 1.3406945 | 98.15331 |
| | BE | 1.3638826 | 92.150718 |
| | BCS | 1.34938 | 94.1988 |
| MOGOA | BC | 1.2822754 | 114.45838 |
| | BE | 1.3853273 | 93.275504 |
| | BCS | 1.29844 | 108.6742 |

Bold values have the best performance; best cost (BC); best emission (BE); best compromise solution (BCS).

The power outputs of the generating units are tabulated in Table 7 and the 24 h total fuel cost value and the value of the total emissions were USD 129,200.076 and 98.1415 Ib, respectively. It can be proved from Table 7 that the total generated power was equal to the summation of $P_L$ and $P_D$ and that all the inequality constraints were fulfilled, which shows the efficacy of the MSSA. The output power of each generator, total power, load demand, and power loss are shown in Figure 10.

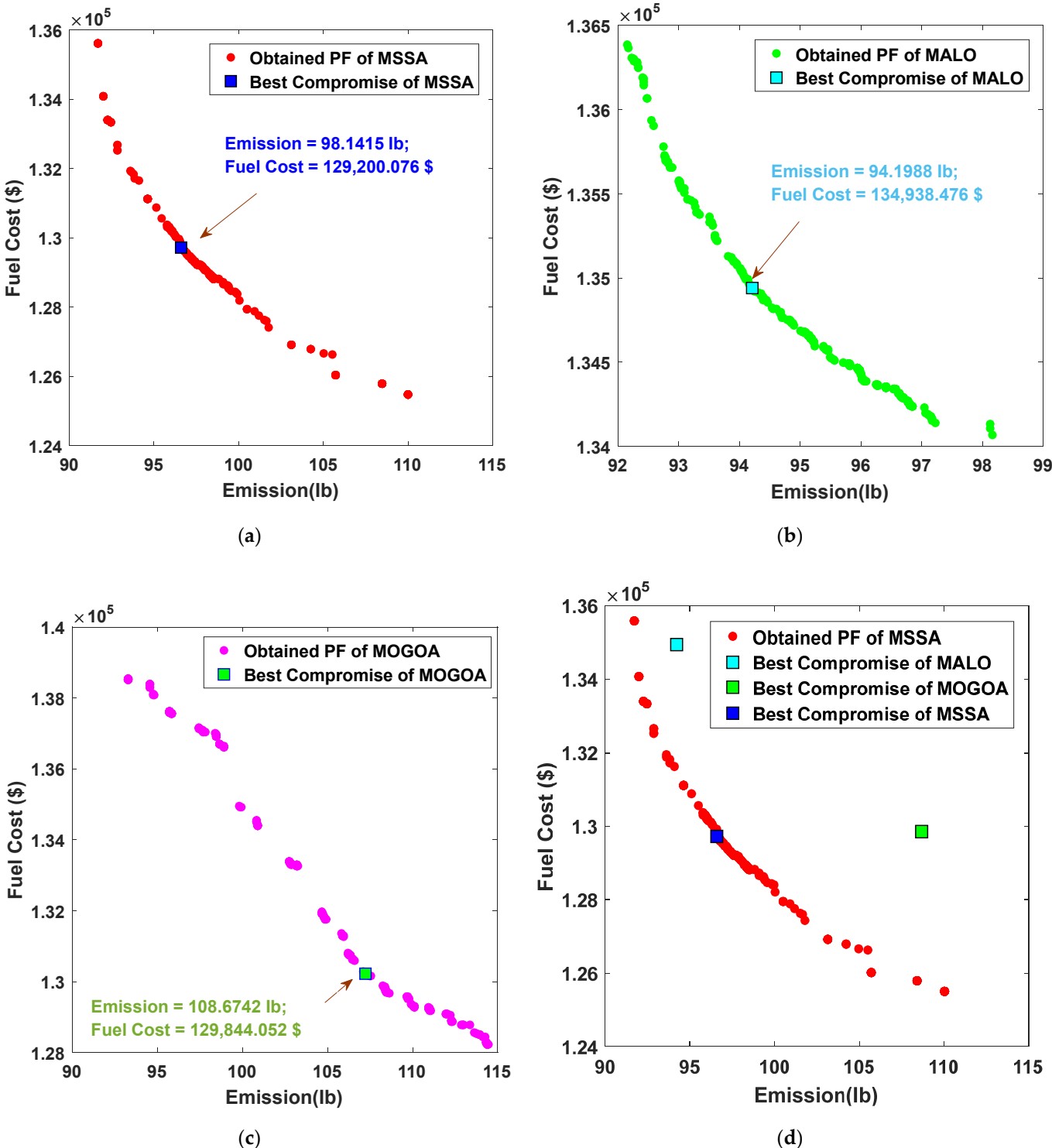

**Figure 9.** Pareto Front of the third case with power loss obtained by (**a**) MSSA; (**b**) MALO; (**c**) MOGOA; (**d**) MSSA and comparison of optimum solutions.

**Table 7.** The BCS of the third case attained using MSSA with transmission losses (p.u.).

| Hour | $P_1$ | $P_2$ | $P_3$ | $P_4$ | $P_5$ | $P_6$ | $P_7$ | $P_8$ | $P_9$ | $P_{10}$ | $P_{11}$ | $P_{12}$ | $P_{13}$ | $P_{14}$ | $P_L$ |
|------|-------|-------|-------|-------|-------|-------|-------|-------|-------|----------|----------|----------|----------|----------|-------|
| 1 | 1.8448 | 1.0056 | 0.7031 | 0.8494 | 0.6978 | 0.7539 | 0.6777 | 0.6474 | 0.7207 | 0.8242 | 0.7970 | 1.0141 | 0.6891 | 0.6274 | 1.8522 |
| 2 | 1.8686 | 1.5590 | 0.7619 | 0.8887 | 0.8204 | 0.7799 | 0.8150 | 0.6967 | 0.8426 | 1.0303 | 1.0040 | 1.3621 | 0.7007 | 0.6837 | 1.8136 |
| 3 | 1.9159 | 1.4781 | 0.7493 | 0.8146 | 0.8097 | 0.7498 | 0.7031 | 0.6984 | 0.9535 | 0.8321 | 0.8117 | 0.8954 | 0.6964 | 0.6315 | 1.7394 |
| 4 | 1.5969 | 1.2879 | 0.6443 | 0.7765 | 0.7695 | 0.7960 | 0.6510 | 0.6319 | 0.6807 | 0.7502 | 0.8043 | 1.1795 | 0.7743 | 0.6046 | 1.9476 |
| 5 | 1.5783 | 1.2570 | 0.8429 | 0.7314 | 0.7314 | 0.7883 | 0.7075 | 0.6977 | 0.7986 | 0.8820 | 0.8005 | 1.0211 | 0.6197 | 0.6664 | 1.9229 |
| 6 | 1.7012 | 1.3836 | 0.8931 | 1.0650 | 0.7281 | 0.8062 | 0.8671 | 0.8092 | 1.0783 | 0.9772 | 0.9998 | 1.1138 | 0.7438 | 0.6578 | 1.8242 |
| 7 | 1.8590 | 1.6225 | 1.0377 | 1.0833 | 1.0213 | 1.0505 | 0.8526 | 0.7107 | 1.0057 | 1.2990 | 1.0806 | 1.1119 | 0.8145 | 0.6702 | 1.7194 |
| 8 | 2.2770 | 1.7767 | 0.8949 | 0.9837 | 0.9789 | 1.0381 | 0.7910 | 0.8754 | 0.9454 | 1.3403 | 1.0184 | 1.2218 | 0.8397 | 0.7222 | 1.6034 |
| 9 | 1.9798 | 2.1283 | 0.8953 | 1.0316 | 0.8775 | 0.9845 | 0.9566 | 0.8269 | 1.2213 | 1.3621 | 1.1799 | 1.5283 | 0.8622 | 0.7757 | 1.7100 |
| 10 | 2.2102 | 1.7363 | 0.9107 | 0.9018 | 0.9547 | 0.8503 | 0.8137 | 1.0557 | 0.9148 | 1.2187 | 1.0422 | 1.3154 | 0.8075 | 0.7241 | 1.6560 |
| 11 | 2.2647 | 1.8614 | 0.8648 | 1.1936 | 1.0429 | 0.8684 | 1.0178 | 0.9309 | 1.0312 | 1.1771 | 1.0551 | 1.5499 | 1.0144 | 0.7607 | 1.6330 |
| 12 | 2.2892 | 1.9223 | 1.2679 | 1.5490 | 0.9783 | 1.1875 | 0.8261 | 0.9638 | 1.2831 | 1.2737 | 1.0947 | 1.6729 | 0.9701 | 0.7903 | 1.5689 |
| 13 | 2.1056 | 1.9158 | 1.0940 | 1.1415 | 0.9491 | 1.2375 | 0.8647 | 0.7604 | 1.0287 | 1.1263 | 1.3653 | 1.4299 | 0.9939 | 0.7363 | 1.6489 |
| 14 | 1.9923 | 1.7601 | 0.9725 | 1.0530 | 1.1690 | 0.9384 | 0.8627 | 0.8816 | 0.8881 | 1.3040 | 1.1878 | 1.4874 | 0.8726 | 0.6524 | 1.7218 |
| 15 | 2.0599 | 1.5643 | 0.8444 | 0.9778 | 0.9095 | 0.8642 | 0.7872 | 0.7977 | 0.9421 | 1.3365 | 1.1358 | 1.0737 | 0.8018 | 0.6860 | 1.6808 |
| 16 | 2.1009 | 1.7972 | 1.1459 | 1.0448 | 1.1458 | 0.9029 | 0.8425 | 0.8539 | 1.0513 | 1.2255 | 1.0421 | 1.3444 | 1.0195 | 0.7369 | 1.6536 |
| 17 | 2.1883 | 2.0192 | 0.9134 | 1.0916 | 1.1254 | 1.0463 | 0.9255 | 0.8499 | 0.9232 | 0.9825 | 1.0277 | 1.3759 | 0.9342 | 0.7155 | 1.6186 |
| 18 | 2.2892 | 2.1489 | 1.0146 | 1.1794 | 0.8802 | 1.1092 | 0.8609 | 0.7866 | 1.0821 | 1.1133 | 1.3164 | 1.4556 | 0.8095 | 0.7232 | 1.5690 |
| 19 | 2.1990 | 1.9796 | 0.8941 | 0.9135 | 1.0583 | 1.0535 | 0.8683 | 0.8093 | 1.0409 | 1.1729 | 1.2207 | 1.6510 | 0.8146 | 0.7100 | 1.6856 |
| 20 | 1.9954 | 1.6575 | 0.9303 | 1.0157 | 0.9446 | 1.1513 | 1.0201 | 0.8893 | 1.1961 | 1.1178 | 1.4577 | 1.3617 | 0.7764 | 0.7405 | 1.7543 |
| 21 | 2.2082 | 1.6239 | 0.8441 | 0.8003 | 0.9282 | 1.0949 | 0.7419 | 0.7442 | 0.9581 | 0.9384 | 1.0039 | 1.4912 | 0.7182 | 0.7269 | 1.7224 |
| 22 | 1.7212 | 1.9025 | 0.8127 | 0.8846 | 1.0127 | 0.8447 | 0.7468 | 0.7491 | 0.8413 | 1.1173 | 0.9040 | 1.2863 | 0.8112 | 0.7732 | 1.8075 |
| 23 | 1.9478 | 1.6296 | 0.8620 | 0.9937 | 0.8662 | 0.8004 | 0.7688 | 0.6997 | 0.9438 | 1.2162 | 0.9303 | 1.2251 | 0.6933 | 0.6424 | 1.7192 |
| 24 | 1.7457 | 1.8476 | 0.8600 | 0.8294 | 0.8312 | 0.9356 | 0.7420 | 0.7182 | 0.8128 | 1.1049 | 0.7625 | 1.1912 | 0.7391 | 0.6600 | 1.7803 |

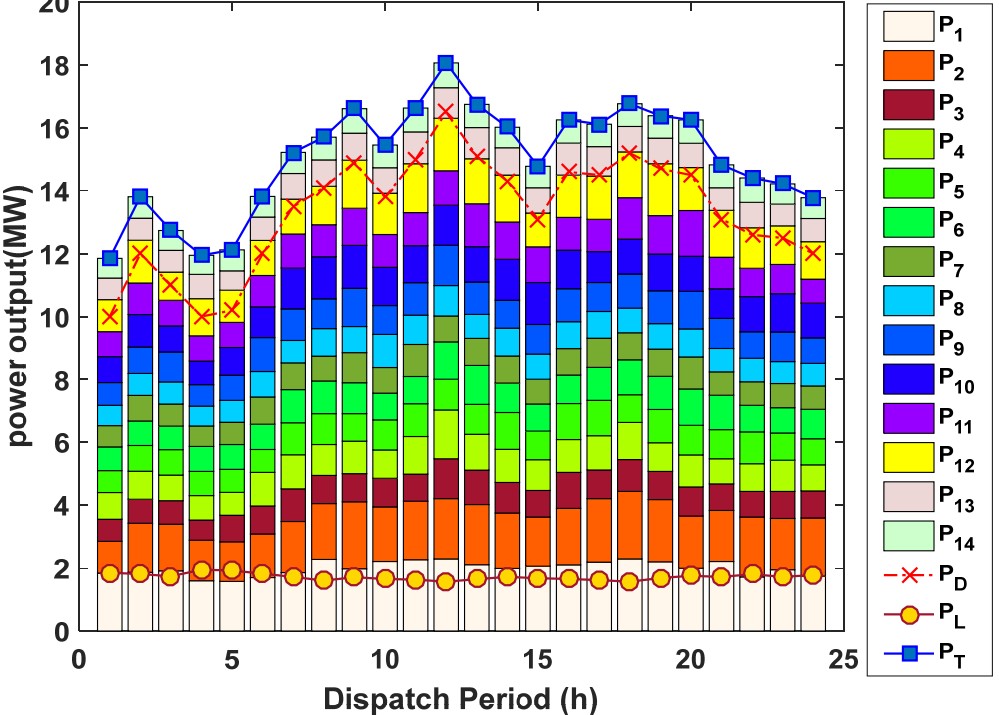

**Figure 10.** Constraints check for the BCS of the third case using MSSA.

The hypervolume (HV) indicator is a set measure used in multi-objective optimization techniques to evaluate the performance of search algorithms. The hypervolume metric is defined as the volume covered by the assessed Pareto Front $Y = (y^{(1)}, y^{(2)}, \ldots, y^{(N)})$ based on a given reference point $t = (t_1, t_2, t_3)$. Therefore, the $HV(Y, t)$ is then defined as [43]:

$$HV(Y,t) = \left(t_1 - y_1^{(1)}\right)\left(t_2 - y_2^{(1)}\right)\left(t_3 - y_3^{(1)}\right) + \sum_{i=2}^{N}\left(t_1 - y_1^{(i)}\right)\left(y_2^{(i-1)} - y_2^{(i)}\right)\left(y_3^{(i-1)} - y_3^{(1)}\right) \tag{13}$$

As depicted in Table 8, the HV of the MSSA outperformed that of the other two techniques in terms of the best, worst, average, and std dev. values, which proves that the technique can achieve ND solutions with a much better capability, consistency, and variety for the three cases compared to the other techniques.

**Table 8.** HV indicator values of results obtained for the three cases.

| Case No. | | MSSA | MALO | MOGOA |
|---|---|---|---|---|
| Case 1 | Best | **0.807653** | 0.636744 | 0.646386 |
| | Worst | **0.789138** | 0.171571 | 0.185795 |
| | Mean | **0.800696** | 0.39641 | 0.438754 |
| | Std dev. | **0.006691** | 0.139975 | 0.14251 |
| Case 2 | Best | **0.665428** | 0.1914 | 0.617644 |
| | Worst | **0.08684** | 0 | 0.049874 |
| | Mean | 0.3104 | 0.07108 | **0.394762** |
| | Std dev. | 0.195818 | **0.080641** | 0.21163 |
| Case 3 | Best | 0.634702 | **0.704149** | 0.578087 |
| | Worst | **0.584107** | 0.427635 | 0.515367 |
| | Mean | **0.610866** | 0.554619 | 0.542181 |
| | Std dev. | **0.015048** | 0.095316 | 0.020538 |

Bold values have the best performance.

## 5. Conclusions

In this article, a DEED model was presented. In this DEED model, the fuel cost and total pollutant emissions were optimized as incompatible objectives through a definite dispatch time. The non-linear aspects of the generating units' ramp rate bounds, VPE, variation of the load demand, and transmission loss were considered. An MSSA was suggested to solve the DEED problem. By testing three cases of three power systems, the obtained results confirmed that the DEED problem was proficiently solved using the MSSA and that a group of well and widely distributed Pareto optimal solutions was acquired rapidly. A comparison of the results with algorithms suggested in related publications showed that the MSSA had better effectiveness and better prospective applications. The study included a comparison with other recent algorithms, including the multi-objective antlion optimizer (MALO), multi-objective grasshopper optimization algorithm (MOGOA), and other published multi-objective algorithms. Additionally, the Pareto Fronts attained for the assessed technique demonstrated the supremacy of the solutions achieved using the MSSA, which indicated the efficacy and applicability of the model in solving complicated MOPs definite to power systems' operations. In addition, all the Pareto Front curves confirmed that the proposed MSSA can give well-disseminated solutions with the fuel cost and emission functions. It was confirmed that the proposed MSSA can achieve well-distributed and high-quality Pareto optimal solutions for the DEED problem and has the possibility to achieve the optimum solution for the MOPs of other power systems. In terms of the model's additional improvement, future studies on the development of the MSSA for solving the power systems' problems including the renewable energy sources in the suggested power system such as wind power and solar power are suggested as, due to their randomness, it will be more complicated to optimize the solution. Therefore, it is necessary to explore a more complete model and to develop the MSSA optimization algorithm to enhance its strength.

**Author Contributions:** Data curation, M.H.H.; Formal analysis, M.H.H. and S.K.; Funding acquisition, M.F.E.-N.; Investigation, M.F.E.-N.; Methodology, M.H.H. and J.L.D.-G.; Project administration, M.F.E.-N.; Resources, S.K. and J.L.D.-G.; Software, J.L.D.-G.; Supervision, S.K.; Visualization, M.F.E.-N.; Writing–original draft, M.H.H.; Writing–review & editing, S.K., J.L.D.-G. and M.F.E.-N. All authors have read and agreed to the published version of the manuscript.

**Funding:** This research was funded by Prince Sattam Bin Abdulaziz University, grant number IF-PSAU-2021/01/18921.

**Institutional Review Board Statement:** Not applicable.

**Informed Consent Statement:** Not applicable.

**Data Availability Statement:** Not applicable.

**Acknowledgments:** The authors extend their appreciation to the Deputyship for Research and Innovation, Ministry of Education in Saudi Arabia for funding this research work through the project number (IF-PSAU-2021/01/18921).

**Conflicts of Interest:** The authors declare no conflict of interest.

## Abbreviations

| | |
|---|---|
| DEED | Dynamic Economic and Emission Dispatch |
| FDM | Fuzzy decision-making |
| MSSA | Multi-objective Salp Swarm Algorithm |
| ITSA | Improved tunicate swarm algorithm |
| MALO | Multi-objective ant lion optimizer |
| MOGOA | Multi-objective grasshopper optimization algorithm |
| EED | Economic and Emission Dispatch |
| SSA | Salp Swarm Algorithm |
| ODEA | The opposition-based differential evolution algorithm |
| PSO | Particle swarm optimizer |
| MODE | Multi-objective Differential Evolution |
| CRO | Chemical Reaction Optimization |
| SQP | Sequential quadratic programming |
| EPFSA | Evolutionary programming-based fuzzy satisfying algorithm |
| ELD | Economic Load Dispatch |
| ND | Non-dominated |
| BCS | Best Compromise Solutions |
| VPE | Valve point effect |
| NSGA-II | Non-dominated Sorting Genetic Algorithm-II |
| MOP | Multi-objective optimization problem |
| DED | Dynamic Economic Dispatch |
| IBFA | Improved bacterial foraging algorithm |
| PSOCS | Particle swarm optimization algorithm with a clone selection |
| DE | Differential evolution |
| MAMODE | Modified Adaptive Multi-objective Differential Evolution |
| GSOMP | Group Search Optimizer with Multiple Producers |
| RWS | Roulette wheel selection |
| MONNDE | Multi-objective Neural Networks trained using Differential Evolution |

## Appendix A

**Table A1.** The load demands (MW) of the six-generator test system [29].

| Hour | $P_D$ | Hour | $P_D$ | Hour | $P_D$ |
|---|---|---|---|---|---|
| 1 | 3.25 | 9 | 5.45 | 17 | 5.15 |
| 2 | 3.90 | 10 | 5.20 | 18 | 5.75 |
| 3 | 3.50 | 11 | 5.50 | 19 | 5.25 |
| 4 | 3.00 | 12 | 5.75 | 20 | 5.25 |
| 5 | 3.35 | 13 | 5.25 | 21 | 4.55 |
| 6 | 4.00 | 14 | 5.15 | 22 | 4.25 |
| 7 | 4.75 | 15 | 4.75 | 23 | 4.25 |
| 8 | 5.05 | 16 | 5.30 | 24 | 4.00 |

**Table A2.** Generator and emission coefficients of the six-generator test system [44].

| Unit | $P_{Gi}^{max}$ | $P_{Gi}^{min}$ | $a_i$(USD/h) | $b_i$(USD/MWh) | $c_i$(USD/(MW)$^2$) | $\alpha_i$ | $\beta_i$ | $\gamma_i$ | $\eta_i$ | $\delta_i$ |
|---|---|---|---|---|---|---|---|---|---|---|
| P$_{G1}$ | 150 | 5 | 10 | 200 | 100 | 4.091 | −5.543 | 6.49 | $2.00 \times 10^{-4}$ | 2.857 |
| P$_{G2}$ | 150 | 5 | 10 | 150 | 120 | 2.543 | −6.047 | 5.638 | $5.00 \times 10^{-4}$ | 3.333 |
| P$_{G3}$ | 150 | 5 | 20 | 180 | 40 | 4.258 | −5.094 | 4.586 | $1.00 \times 10^{-6}$ | 8 |
| P$_{G4}$ | 150 | 5 | 10 | 100 | 60 | 5.326 | −3.55 | 3.38 | $2.00 \times 10^{-3}$ | 2 |
| P$_{G5}$ | 150 | 5 | 20 | 180 | 40 | 4.258 | −5.094 | 4.586 | $1.00 \times 10^{-6}$ | 8 |
| P$_{G6}$ | 150 | 5 | 10 | 150 | 100 | 6.131 | −5.555 | 5.151 | $1.00 \times 10^{-5}$ | 6.667 |

## Appendix B

**Table A3.** The load demands (MW) of the standard 39-bus ten-unit test system [34].

| Hour | P$_D$ | Hour | P$_D$ | Hour | P$_D$ |
|---|---|---|---|---|---|
| 1 | 1036 | 9 | 1924 | 17 | 1480 |
| 2 | 1110 | 10 | 2022 | 18 | 1628 |
| 3 | 1258 | 11 | 2106 | 19 | 1776 |
| 4 | 1406 | 12 | 2150 | 20 | 1972 |
| 5 | 1480 | 13 | 2072 | 21 | 1924 |
| 6 | 1628 | 14 | 1924 | 22 | 1628 |
| 7 | 1702 | 15 | 1776 | 23 | 1332 |
| 8 | 1776 | 16 | 1554 | 24 | 1184 |

**Table A4.** Generator coefficients of the standard 39-bus 10-unit test system [45].

| Unit | $P_{Gi}^{min}$ | $P_{Gi}^{max}$ | $a_i$(USD/h) | $b_i$(USD/MWh) | $c_i$(USD/(MW)$^2$) | $d_i$(USD/h) | $e_i$(rad/MW) |
|---|---|---|---|---|---|---|---|
| P$_{G1}$ | 150 | 470 | 786.7988 | 38.5379 | 0.1524 | 450 | 0.041 |
| P$_{G2}$ | 135 | 470 | 451.3251 | 46.1591 | 0.1058 | 600 | 0.036 |
| P$_{G3}$ | 73 | 340 | 1049.998 | 40.3965 | 0.028 | 320 | 0.028 |
| P$_{G4}$ | 60 | 300 | 1243.531 | 38.3055 | 0.0354 | 260 | 0.052 |
| P$_{G5}$ | 73 | 243 | 1658.57 | 36.3278 | 0.0211 | 280 | 0.063 |
| P$_{G6}$ | 57 | 160 | 1356.659 | 38.2704 | 0.0179 | 310 | 0.048 |
| P$_{G7}$ | 20 | 130 | 1450.705 | 36.5104 | 0.0121 | 300 | 0.086 |
| P$_{G8}$ | 47 | 120 | 1450.705 | 36.5104 | 0.0121 | 340 | 0.082 |
| P$_{G9}$ | 20 | 80 | 1455.606 | 39.5804 | 0.109 | 270 | 0.098 |
| P$_{G10}$ | 10 | 55 | 1469.403 | 40.5407 | 0.1295 | 380 | 0.094 |

**Table A5.** Emission coefficients and ramp rate of the standard 39-bus 10-unit test system [45].

| Unit | $\alpha_i$ | $\beta_i$ | $\gamma_i$ | $\eta_i$ | $\delta_i$ | $UR_i$ | $DR_i$ |
|---|---|---|---|---|---|---|---|
| P$_1$ | 103.3908 | 2.4444 | 0.0312 | 0.5035 | 0.0207 | 80 | 80 |
| P$_2$ | 103.3908 | 2.4444 | 0.0312 | 0.5035 | 0.0207 | 80 | 80 |
| P$_3$ | 300.391 | 4.0695 | 0.0509 | 0.4968 | 0.0202 | 80 | 80 |
| P$_4$ | 300.391 | 4.0695 | 0.0509 | 0.4968 | 0.0202 | 50 | 50 |
| P$_5$ | 320.0006 | 3.8132 | 0.0344 | 0.4972 | 0.02 | 50 | 50 |
| P$_6$ | 320.0006 | 3.8132 | 0.0344 | 0.4972 | 0.02 | 50 | 50 |
| P$_7$ | 330.0056 | 3.9023 | 0.0465 | 0.5163 | 0.0214 | 50 | 30 |
| P$_8$ | 330.0056 | 3.9023 | 0.0465 | 0.5163 | 0.0214 | 30 | 30 |
| P$_9$ | 350.0056 | 3.9524 | 0.0465 | 0.5475 | 0.0234 | 30 | 30 |
| P$_{10}$ | 360.0012 | 3.9864 | 0.047 | 0.5475 | 0.0234 | 30 | 30 |

**Table A6.** The B-coefficients of the standard 39-bus 10-unit test system [45].

| $B_{ij}$ | 0.000049 | 0.000014 | 0.000015 | 0.000015 | 0.000016 | 0.000017 | 0.000017 | 0.000018 | 0.000019 | 0.00002 |
|---|---|---|---|---|---|---|---|---|---|---|
| | 0.000014 | 0.000045 | 0.000016 | 0.000016 | 0.000017 | 0.000015 | 0.000015 | 0.000016 | 0.000018 | 0.000018 |
| | 0.000015 | 0.000016 | 0.000039 | 0.00001 | 0.000012 | 0.000012 | 0.000014 | 0.000014 | 0.000016 | 0.000016 |
| | 0.000015 | 0.000016 | 0.00001 | 0.00004 | 0.000014 | 0.00001 | 0.000011 | 0.000012 | 0.000014 | 0.000015 |
| | 0.000016 | 0.000017 | 0.000012 | 0.000014 | 0.000035 | 0.000011 | 0.000013 | 0.000013 | 0.000015 | 0.000016 |
| | 0.000017 | 0.000015 | 0.000012 | 0.00001 | 0.000011 | 0.000036 | 0.000012 | 0.000012 | 0.000014 | 0.000015 |
| | 0.000017 | 0.000015 | 0.000014 | 0.000011 | 0.000013 | 0.000012 | 0.000038 | 0.000016 | 0.000016 | 0.000018 |
| | 0.000018 | 0.000016 | 0.000014 | 0.000012 | 0.000013 | 0.000012 | 0.000016 | 0.00004 | 0.000015 | 0.000016 |
| | 0.000019 | 0.000018 | 0.000016 | 0.000014 | 0.000015 | 0.000014 | 0.000016 | 0.000015 | 0.000042 | 0.000019 |
| | 0.00002 | 0.000018 | 0.000016 | 0.000015 | 0.00016 | 0.000015 | 0.000018 | 0.000016 | 0.000019 | 0.000044 |
| $B_{i0}$ | 0 | 0 | 0 | 0 | 0 | 0 | 0 | 0 | 0 | 0 |
| $B_{00}$ | 0 | | | | | | | | | |

**Appendix C**

**Table A7.** The load demands (MW) of the IEEE 118-bus 14-generator test system [29].

| Hour | $P_D$ | Hour | $P_D$ | Hour | $P_D$ |
|---|---|---|---|---|---|
| 1 | 10.0 | 9 | 14.9 | 17 | 14.5 |
| 2 | 12.0 | 10 | 13.8 | 18 | 15.2 |
| 3 | 11.0 | 11 | 15.0 | 19 | 14.7 |
| 4 | 10.0 | 12 | 16.5 | 20 | 14.5 |
| 5 | 10.2 | 13 | 15.1 | 21 | 13.1 |
| 6 | 12.0 | 14 | 14.3 | 22 | 12.6 |
| 7 | 13.5 | 15 | 13.1 | 23 | 12.5 |
| 8 | 14.1 | 16 | 14.6 | 24 | 12.0 |

**Table A8.** Generator and emission coefficients of the IEEE 118-bus 14-generator test system [46].

| Unit | $P_{Gi}^{max}$ | $P_{Gi}^{min}$ | $a_i$(USD/h) | $b_i$(USD/MWh) | $c_i$(USD/(MW)$^2$) | $\alpha_i$ | $\beta_i$ | $\gamma_i$ |
|---|---|---|---|---|---|---|---|---|
| $P_{G1}$ | 300 | 50 | 150 | 189 | 0.5 | 0.016 | −1.5 | 23.333 |
| $P_{G2}$ | 300 | 50 | 115 | 200 | 0.55 | 0.031 | −1.82 | 21.022 |
| $P_{G3}$ | 300 | 50 | 40 | 350 | 0.6 | 0.013 | −1.249 | 22.05 |
| $P_{G4}$ | 300 | 50 | 122 | 315 | 0.5 | 0.012 | −1.355 | 22.983 |
| $P_{G5}$ | 300 | 50 | 125 | 305 | 0.5 | 0.02 | −1.9 | 21.313 |
| $P_{G6}$ | 300 | 50 | 70 | 275 | 0.7 | 0.007 | 0.805 | 21.9 |
| $P_{G7}$ | 300 | 50 | 70 | 345 | 0.7 | 0.015 | −1.401 | 23.001 |
| $P_{G8}$ | 300 | 50 | 70 | 345 | 0.7 | 0.018 | −1.8 | 24.003 |
| $P_{G9}$ | 300 | 50 | 130 | 245 | 0.5 | 0.019 | −2 | 25.121 |
| $P_{G10}$ | 300 | 50 | 130 | 245 | 0.5 | 0.012 | −1.36 | 22.99 |
| $P_{G11}$ | 300 | 50 | 135 | 235 | 0.55 | 0.033 | −2.1 | 27.01 |
| $P_{G12}$ | 300 | 50 | 200 | 130 | 0.45 | 0.018 | −1.8 | 25.101 |
| $P_{G13}$ | 300 | 50 | 70 | 345 | 0.7 | 0.018 | −1.81 | 24.313 |
| $P_{G14}$ | 300 | 50 | 45 | 389 | 0.6 | 0.03 | −1.921 | 27.119 |

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
