# Peer review of "MSSA-DEED: A Multi-Objective Salp Swarm Algorithm for Solving Dynamic Economic Emission Dispatch Problems"

_sustainability, doi:10.3390/su14159785_

Round 1
Reviewer 1 Report
The paper concerns with very important topic, which receive the researchers interests. The authors should mention the reasons for choosing the suggested techniques clearly in the abstract. The Figures 3 and 5 must be cited. The introduction needs more references. The conclusions is too long, it should be reduced and concentrate on the article results.
Author Response
Dear Editor and Reviewers
Thank you very much for your vital comments on the manuscript entitled “MSSA- DEED: A Multi-objective Salp Swarm Algorithm for Solving Dynamic Economic Emission Dispatch Problems”, ID: 1809463. We have made a thorough revision of the manuscript based on the comments of the editor and reviewers. We appreciate the time and effort dedicated to providing feedback on the manuscript. These changes are highlighted in the updated manuscript. A point-by-point response to reviewers' comments and concerns is written in Red and highlighted in YELLOW in the updated manuscript.
Reviewer #1:
- The paper concerns with very important topic, which receive the researchers interests. The authors should mention the reasons for choosing the suggested techniques clearly in the abstract.
Response: We appreciate the reviewer’s comment. The Abstract has been updated as follows:
Salp Swarm Algorithm (SSA) is a well-established metaheuristic that was inspired by the foraging behavior of salps in deep oceans and has proved to be beneficial in estimating global optima for many optimization problems. The objective of this article is to evaluate the performance of the multi-objective Salp Swarm Algorithm (MSSA) for obtaining the optimal dispatching schemes.
- The Figures 3 and 5 must be cited.
Response: We appreciate the reviewer’s comment. These figures have been cited as follows:
The IEEE 30-bus 6-generator (Case 1) test system is displayed in Figure 3. The single line diagram of the 10-generator (Case 2) test system is displayed in Figure 4.
- The introduction needs more references.
Response: We appreciate the reviewer’s comment. The introduction has been updated in the revised paper.
- The conclusions is too long, it should be reduced and concentrate on the article results.
Response: We appreciate the reviewer’s comment. The conclusion has been updated in the revised paper.
Reviewer 2 Report
The application of the Multi-objective Salp Swarm Algorithm for Dynamic Economic Emission Dispatch has been covered in this study. The article is skillfully written. It has compared the outcomes with a few already available and previously published findings. My thoughts on this work are as follows:
1) The research compared the outcomes with certain well-known algorithms such as MALO and MOGOA. However, those algorithms were used in considerably more complicated situations, such as non-linear loads, integrated systems for renewable energy, or smart grids. But a simpler scenario was thought of in this paper. How does the author ensure that MSSA works for non-linear loads or smart grids? If the authors could include these elements in their study, that would be fantastic.
2) What issues could occur when integrating renewable energy sources like wind energy into a system that uses converters and/or variable speed drives?
3) What is the system's THD for the three cases?
4) It is possible to lower the similarity index.
Author Response
Dear Editor and Reviewers
Thank you very much for your vital comments on the manuscript entitled “MSSA- DEED: A Multi-objective Salp Swarm Algorithm for Solving Dynamic Economic Emission Dispatch Problems”, ID: 1809463. We have made a thorough revision of the manuscript based on the comments of the editor and reviewers. We appreciate the time and effort dedicated to providing feedback on the manuscript. These changes are highlighted in the updated manuscript. A point-by-point response to reviewers' comments and concerns is written in Red and highlighted in YELLOW in the updated manuscript.
Reviewer #2:
The application of the Multi-objective Salp Swarm Algorithm for Dynamic Economic Emission Dispatch has been covered in this study. The article is skillfully written. It has compared the outcomes with a few already available and previously published findings. My thoughts on this work are as follows:
- The research compared the outcomes with certain well-known algorithms such as MALO and MOGOA. However, those algorithms were used in considerably more complicated situations, such as non-linear loads, integrated systems for renewable energy, or smart grids. But a simpler scenario was thought of in this paper. How does the author ensure that MSSA works for non-linear loads or smart grids? If the authors could include these elements in their study, that would be fantastic.
Response: We appreciate the reviewer’s comment. In this paper, the DEED problem is a more complex problem compared to several optimization problems because it has different values of the load demand through 24 hours that needs to be scheduled and chose the best values of the power generation through this period.
The DEED issue is a highly dimensional, deeply correlated, nonconvex and nonlinear MOP taking into consideration the incompatible objectives and the operational limitations. When the transmission losses, the changing of the power demand and the POZ were also involved, the DEED model will be more complex.
- What issues could occur when integrating renewable energy sources like wind energy into a system that uses converters and/or variable speed drives?
Response: We appreciate the reviewer’s comment. In this article, the renewable energy sources have not been taken into consideration but it was added in future work as follows:
In terms of the model’s additional improvement, future work studies the development of the MSSA for solving the power systems problems including the renewable energy sources in the suggested power system such as wind power and solar power, due to their randomness; it will be more complicated to optimize the solution. Therefore, it is necessary to explore a more complete model and develop the MSSA optimization algorithm to enhance its strength.
- What is the system's THD for the three cases?
Response: We appreciate your question. The system's THD has not been considered in the current study.
- It is possible to lower the similarity index.
Response: We appreciate the reviewer’s comment. The similarity index has been decreased in the revised paper.
Reviewer 3 Report
The paper is well written. The paper is technically sound and contributes towards original research. The topic is very relevant in today's scenario. The references are adequate.
Author Response
Dear Editor and Reviewers
Thank you very much for your vital comments on the manuscript entitled “MSSA- DEED: A Multi-objective Salp Swarm Algorithm for Solving Dynamic Economic Emission Dispatch Problems”, ID: 1809463. We have made a thorough revision of the manuscript based on the comments of the editor and reviewers. We appreciate the time and effort dedicated to providing feedback on the manuscript. These changes are highlighted in the updated manuscript. A point-by-point response to reviewers' comments and concerns is written in Red and highlighted in YELLOW in the updated manuscript.
Reviewer #3:
The paper is well written. The paper is technically sound and contributes towards original research. The topic is very relevant in today's scenario. The references are adequate.
Response: The authors greatly appreciate the reviewer’s encouraging comment.
Reviewer 4 Report
This paper uses an algorithm called Multi-objective Salp Swarm Algorithm or MSSA to solve a dynamic economic emission dispatch or DEED problem. Overall, the paper is neither well written nor presents any new or exciting research work. It simply regurgitates old material into a new package using fancy acronyms that can be best qualified as a course project. It does not contribute any material that is worth archiving as a journal article. Detailed comments are as follows.
1. The economic dispatch problem and its many variations have been well-studied in the literature. Compared to the existing volume of literature, the literature survey in the paper is insufficient.
2. None of the contributions listed by the authors can be considered as actual contributions. The authors have simply used an existing algorithm to solve a very well-known problem, which is equivalent to a course project. The reviewer does not recognize any technical innovation or modeling contribution in this work. This could have been considered as a contribution had the authors developed the optimization algorithm themselves. In reference to the second contribution listed by the authors, adding a few constraints to a very well-established framework can hardly be labeled as a contribution. Similar logic can be applied to the rest of the points. In terms of the improvement reported:
a. The MSSA method has been compared to only a select few methods. How does it fare when compared with other more popular multi-objective algorithms like genetic algorithm, NSGA-II, etc.?
b. The improvement shown in the results is not significant enough for this work to be archived.
3. A lot of redundant material has been added in the manuscript that has unnecessarily lengthened the manuscript. For instance:
a. Load curves of test systems
b. Very detailed datasets for test systems. A proper reference and a summary would have sufficed. An appendix could be used if the authors think that including the data is really important for their work.
c. Tables 11, 13, 15, etc. The authors should put all these results in a separate appendix section instead of cluttering the main body of work.
4. The overall quality of writing is poor. The manuscript is full of grammatical errors, unnecessarily long sentences, basic sentence construction issues, overuse of transition words, and typos. These characteristics make reading the manuscript a cumbersome activity. Published research work should be written in a manner that is well-structured and easy for the readers to follow.
5. Adjectives like “best” (as used in the abstract to describe DEED) should be used judiciously in scientific papers. It is advised not to use such words without backing them up with sufficient evidence.
6. Way too many acronyms have been used, which the reviewer deems to be unnecessary and a hindrance to the readers.
Author Response
Dear Editor and Reviewers
Thank you very much for your vital comments on the manuscript entitled “MSSA- DEED: A Multi-objective Salp Swarm Algorithm for Solving Dynamic Economic Emission Dispatch Problems”, ID: 1809463. We have made a thorough revision of the manuscript based on the comments of the editor and reviewers. We appreciate the time and effort dedicated to providing feedback on the manuscript. These changes are highlighted in the updated manuscript. A point-by-point response to reviewers' comments and concerns is written in Red and highlighted in YELLOW in the updated manuscript.
This paper uses an algorithm called Multi-objective Salp Swarm Algorithm or MSSA to solve a dynamic economic emission dispatch or DEED problem. Overall, the paper is neither well written nor presents any new or exciting research work. It simply regurgitates old material into a new package using fancy acronyms that can be best qualified as a course project. It does not contribute any material that is worth archiving as a journal article. Detailed comments are as follows.
- The economic dispatch problem and its many variations have been well-studied in the literature. Compared to the existing volume of literature, the literature survey in the paper is insufficient.
Response: We appreciate the reviewer’s comment. The literature survey has been updated in the revised paper.
- None of the contributions listed by the authors can be considered as actual contributions. The authors have simply used an existing algorithm to solve a very well-known problem, which is equivalent to a course project. The reviewer does not recognize any technical innovation or modeling contribution in this work. This could have been considered as a contribution had the authors developed the optimization algorithm themselves. In reference to the second contribution listed by the authors, adding a few constraints to a very well-established framework can hardly be labeled as a contribution. Similar logic can be applied to the rest of the points. In terms of the improvement reported:
- The MSSA method has been compared to only a select few methods. How does it fare when compared with other more popular multi-objective algorithms like genetic algorithm, NSGA-II, etc.?
- The improvement shown in the results is not significant enough for this work to be archived.
Response: We appreciate the reviewer’s comment. The MSSA algorithm is compared to two multi-objective algorithms MALO, MOGOA, as well as six multi-objective algorithms that were used to solve the dynamic economic emission dispatch (DEED) problem such as MAMODE, GSOMP, MOPSO, NSGA-II, RCGA, and MODE.
- A lot of redundant material has been added in the manuscript that has unnecessarily lengthened the manuscript. For instance:
- Load curves of test systems
- Very detailed datasets for test systems. A proper reference and a summary would have sufficed. An appendix could be used if the authors think that including the data is really important for their work.
- Tables 11, 13, 15, etc. The authors should put all these results in a separate appendix section instead of cluttering the main body of work.
Response: We appreciate the reviewer’s comment. The simulation part has been updated in the revised paper.
- The overall quality of writing is poor. The manuscript is full of grammatical errors, unnecessarily long sentences, basic sentence construction issues, overuse of transition words, and typos. These characteristics make reading the manuscript a cumbersome activity. Published research work should be written in a manner that is well-structured and easy for the readers to follow.
Response: We appreciate the reviewer’s comment. The manuscript has been revised.
- Adjectives like “best” (as used in the abstract to describe DEED) should be used judiciously in scientific papers. It is advised not to use such words without backing them up with sufficient evidence.
Response: We appreciate the reviewer’s comment. It has been edited in the revised manuscript.
- Way too many acronyms have been used, which the reviewer deems to be unnecessary and a hindrance to the readers.
Response: We appreciate the reviewer’s comment. They have been edited in the revised paper.
Round 2
Reviewer 4 Report
I appreciate the efforts made by the authors to improve their manuscript. However, I still don't recognize the contribution of this work to be of archival quality. I reiterate the points that I made in my previous review: there is no contribution in terms of modeling or technical innovation. The revised version has not improved on these particular areas. The authors have simply used an existing algorithm to solve a very well-known problem with marginal improvements in results, which is not a strong enough technical contribution in my opinion.